# Political Islam: A 40 Year Retrospective

## Nader Hashemi

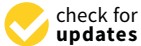

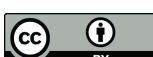

Josef Korbel School of International Studies, University of Denver, Denver, CO 80208, USA; nader.hashemi@du.edu

**Abstract:** The year 2020 roughly corresponds with the 40th anniversary of the rise of political Islam on the world stage. This topic has generated controversy about its impact on Muslims societies and international affairs more broadly, including how governments should respond to this socio-political phenomenon. This article has modest aims. It seeks to reflect on the broad theme of political Islam four decades after it first captured global headlines by critically examining two separate but interrelated controversies. The first theme is political Islam's acquisition of state power. Specifically, how have the various experiments of Islamism in power effected the popularity, prestige, and future trajectory of political Islam? Secondly, the theme of political Islam and violence is examined. In this section, I interrogate the claim that mainstream political Islam acts as a "gateway drug" to radical extremism in the form of Al Qaeda or ISIS. This thesis gained popularity in recent years, yet its validity is open to question and should be subjected to further scrutiny and analysis. I examine these questions in this article.

**Keywords:** political Islam; Islamism; Islamic fundamentalism; Middle East; Islamic world; Muslim Brotherhood

## 1. Introduction

The year 2020 roughly coincides with the 40th anniversary of the rise of political Islam.[1] While this trend in Muslim politics has deeper historical and intellectual roots, it was approximately four decades ago that this subject emerged from seeming obscurity to capture global attention. Several events overlapped that contributed to this development: the 1979 Islamic Revolution Iran and the capture of the American embassy in Tehran, the seizure of the Grand Mosque in Mecca by Saudi militants, the Soviet invasion of Afghanistan and the rise of the Afghan Mujahideen, Muhammad Zia-ul-Haq's Islamization program in Pakistan, and the growing prominence of the Muslim Brotherhood movement in societies throughout the Arab world.

It was during this period that the prominent historian of Islam, Bernard Lewis, weighed in on this topic. Attending a colloquium on 23 June 1980 at Tel Aviv University, he predicted that notwithstanding these dramatic events, the "Islamic resurgence has reached its peak and that from now onwards it [will] probably decline rather than ascend."[2] The past 40 years have proven Bernard Lewis wrong and not for the last time.[3]

For four decades, various manifestations of political Islam have dominated global affairs. In late 2020, after a series of terrorist attacks in France and Austria, governments in both countries announced a formal ban on "political Islam." President Emmanuel Macron

---

1  I am utilizing Andrew March's definition that "political Islam should be understood in the broadest sense possible as the range of modern political movements, ideological trends, and state-directed policies concerned with giving Islam an authoritative status in political life." See (March 2015). My focus is on the mainstream expressions of political Islam in the Middle East and North Africa. The terms "political Islam", "Islamism", and "Islamic fundamentalism" are often used interchangably in public and scholarly discussion. I eschew the term "Islamic fundamentalism" as it is more accusatory than descriptive. For a discussion, see (Hashemi 2009).

2  (Ben Dor 1983).

3  Bernard Lewis was a key advisor to the Bush Administration in the lead up to the 2003 Iraq war. He famously predicted that public opinion in Iraq and Iran was extremely pro-American and that both peoples would rejoice if the United States liberated them. See (Buruma 2004).

gave French Muslim leaders a 15-day ultimatum to work with the Ministry of Interior on a "charte des valeurs républicaines" (charter of republican values) that includes as a key principle "the rejection of political Islam." In Austria, Chancellor Sebastian Kurz proclaimed an identical goal. "We will create a criminal offense called 'political Islam' in order to be able to take action against those who are not terrorists themselves but who create the breeding ground for such."[4]

These efforts recall various unsuccessful attempts in the United States to officially ban the Muslim Brotherhood by declaring it a foreign terrorist organization.[5] Other governments have pursued similar efforts. A key takeaway from these developments is that after 40 years of grappling with this subject, outside a small group of academic experts, political Islam remains poorly understood. In Western popular culture, in intellectual circles, and among policy makers, the term political Islam, Islamism, or the Muslim Brotherhood has deep pejorative associations. The concept of political Islam is reductively and often erroneously equated with violence, fanaticism, authoritarianism and terrorism.[6]

This article has modest aims. It seeks to reflect on the broad theme of political Islam on its 40th anniversary by critically examining two separate but interrelated controversies that this topic has generated. The first theme is political Islam's acquisition of state power. Specifically, how have the various experiments of Islamism in power effected the popularity, prestige and future trajectory of this socio-political movement? Secondly, the theme of political Islam and violence will be discussed. In this section, I interrogate the claim that mainstream political Islam acts as a "gateway" drug to radical extremism in the form of Al Qaeda or ISIS. This thesis gained popularity in recent years, yet the question remains as to whether it has any validity? I examine these questions in the pages below.

## 2. Political Islam in Power

Generalizations about political Islam are almost always misleading. Understanding the local and national context is essential if one seeks a firm grasp of the dynamics of religious politics in Muslim societies.[7] Nonetheless, there are a set of overarching truths that apply to this subject. First, political Islam emerged in response to the decline of Muslim civilization during the 19th and 20th centuries. This is deeply connected to the impact and legacy of Euro-American imperialism and the failure of post-colonial regimes. Second, political Islamists gradually developed a political theory for a just society premised on a critique of secular paradigms and anchored in an argument about Islamic authenticity.[8] Its appeal increased over time in direct proportion to the failure of Muslim ruling elites to advance political and economic development. Third, for most of its history, political Islam was an opposition movement. Forty years ago, political Islam had no direct experience with political power nor any control over state institutions. On this latter point, much has changed during last few decades, resulting in significant consequences for the future trajectory, appeal and orientation of political Islam.

Looking back over the past 40 years, political Islam's experience with political power, stands out as an important development. Morocco, Tunisia, Egypt, Gaza, Sudan, Turkey, Jordan, Kuwait, Iran, Iraq, Afghanistan, and Malaysia have all experienced religious-based political movements contesting political power and, to varying degrees, obtaining control of the state. Admittedly, these experiences are extremely complex, multifaceted, and heterogeneous, and generalizations are hazardous to make. Some Islamist forces have come to power via revolution and military coups/conquest, while others have risen to power via the ballot box often in semi-authoritarian settings.

---

4    (Wesfreid 2020; Lawler 2020).

5    (Ballhaus et al. 2019).

6    (Clooney 2020).

7    (Ayoob and Lussier 2020; Ayubi 1991; Brown 2000; Piscatori and Eickelman 1996; Esposito 1998; Beinin and Stork 1997; Zubaida 1993; Burgat 2021).

8    (Enayat 1982; Rahmena 2005). Also see the reference in footnote 7.

In this context, one general theme stands: the popular prestige of political Islam has been tarnished by its experience with state power. As an opposition movement, it once occupied the high moral ground. Its critique of the secular ideologies, the political status quo, and Western policy toward the Middle East was appealing to many segments of society. To those sympathetic to its narrative, the Islamist vision for a new political order was given the benefit of the doubt. Arguably, this has significantly changed since 1980, specifically due to the interaction and acquisition of state power by political Islamist movements. Islamism writ large has lost its pristine image, its perceived innocence, and some of its luster. This trend has been confirmed by a major 2019 study by the Arab Barometer that noted a significant decline in support for religious parties and leaders.[9]

The various Islamist experiences with political power have forged new attitudes and shaped perceptions about religious politics both within the Islamic world and outside of it. There is now a track record of Islamist behavior and policies than can be judged. There is also list of victims who have suffered from these policies and who now have grievances that did not exist when political Islam was an opposition movement. Unfulfilled expectations, incompetent governance, and promises not kept are also part of this story.

Admittedly, after 40 years, political Islam still has significant support. The mobilization of Muslims around religious identity markers does still take place, and secular alternatives remain weak. It is extremely difficult to measure societal support for religious parties today given current political conditions in the Arab-Islamic world, which are marked by deepening authoritarian repression, expanding censorship, and the absence of free and fair elections. The coronavirus crisis has exacerbated these conditions.[10]

In assessing the political Islamist record in power (for which there is not one but several), it must be acknowledged that the playing field is not level. In many cases, when Islamists won elections, real power still resided in the deep state and behind the political thrones of monarchs and military men. Muslim-based politicians and political parties did not have full autonomy to pursue their own policies and implement their agendas.

Moreover, Islamists in power faced immense challenges regionally and internationally. Threats of regime change and outside financial support directed at removing them from power, were persistent factors that circumscribed their ability to rule. This was most apparent in Egypt during the Arab Spring. While the Muslim Brotherhood won parliamentary and presidential elections after the ouster of Hosni Mubarak, it faced persistent interference from regional powers who collaborated with the military to remove Mohammad Morsi from office.[11]

The waning appeal of political Islam is most apparent in Iran. Four decades of Islamist rule has produced the opposite—societal secularization on a wide scale, especially among young people, in urban areas, and among the middle class. Iranian opposition politics is also decidedly secular in the sense that democratization in Iran now is intimately linked with anti-clericalism and the demand to separate religion from state.[12] As the dissident scholar and theologian Mohsen Kadivar has argued, "the process of secularization hasn't stopped, and there's no way can we stop it."[13] He attributed this directly to the failure of political Islam in Iran.

Senior Iranian leaders have openly acknowledged this problem. On the 40th anniversary of the Islamic Revolution in Iran, the influential hardline cleric, Ayatullah Mesbah-Yazdi, gave an interview where he acknowledged that "Iranians are evading religious teachings and turning to secularism." He admitted that the Supreme Leader was aware of

---

9    (The Economist 2019). Also see (Kurzman and Türkoglu 2015).

10   (Dunn 2020).

11   (Kirkpatrick 2015a).

12   (Mousavi 2011).

13   (Harbin 2012; Hashemi 2018; Arab and Maliki 2020).

this trend. He also added that the problem was so pervasive that "we see traces of secularism even among the country's top officials. This is the case even in the seminaries."[14]

Thirty years of military rule in Sudan, backed by the forces of political Islam, produced similar results. What stands out about Sudan under the leadership of Omar al-Bashir is not only massive corruption, economic mismanagement, and authoritarian rule but also human rights violations on a massive scale. This has resulted in an indictment against Bashir for war crimes, crimes against humanity, and genocide in Darfur by the International Criminal Court. Global isolation and sanctions have followed, not to mention a devastating civil war in the south that led to the partition of the country

A democratic uprising began in Sudan in December 2018. It succeeded in toppling Bashir's dictatorship in April 2019. After months of sustained protests, a three-year transition to democracy was negotiated between the Transitional Military Council and the Forces for Freedom of Change, backed by the international community. The reputation of political Islam in Sudan was thoroughly tarnished by its association with the old order.

In this story, the figure of Hasan al-Turabi looms large. According to Human Rights Watch, not only was he "the power behind the throne" (until he broke with General Bashir), but he was also a prominent theoretician of political Islam.[15] Arguably, Turabi's influence among Sunni Islamists can be placed in the same category as Hasan Al-Banna, Sayyid Qutb, and Abul A'la Maududi. Should a future democratic Sudan emerge and consolidate itself, political Islam will be remembered in Sudanese history books for its close association with the dark days of military dictatorship.

After the 1989 coup that brought Omar al-Bashir and the Islamists to power, the Sudanese philosopher and public intellectual Abdelwahab El-Affendi wrote an important book, *Who Needs an Islamic State*? Forecasting the disaster that was to emerge, he predicted conflict, bloodshed, and a massive suppression of democratic and human rights under an Islamist vision for Sudan. He was specifically critical of the utopian idealism and the apocalyptic fervor of Islamists who claimed they understood God's will and were intent on imposing it on society. Instead of focusing on political power, El-Affendi argued that Muslim activists should concentrate their efforts on promoting ethically grounded Islamic values and virtues.[16] Looking back, he observes that "for most Sudanese, Islamism came to signify corruption, hypocrisy, cruelty and bad faith." He described the 2018–2019 democratic uprising in Sudan as "the first genuinely anti-Islamist country in popular terms."[17]

The other prominent example of political Islam in power is in Afghanistan under the Taliban. Although the Taliban governed for a short time, between 1996–2001, their record of political rule, marked by a strict interpretation and imposition of Islamic law, generated global outrage. They are remembered today for their persecution of women, religious minorities, cultural destruction, and, most tellingly, for harboring Al Qaeda.[18] Across the Muslim world, beyond their own ethnic Pashtun support base, they have few followers, and their steadfast resistance to the America's military presence has not changed this fact. Internationally, Taliban rule is a metaphor for medieval barbarism.

By contrast, political Islamists who have come to power via the ballot enjoy greater legitimacy and popularity than those who have obtained power via revolution, coups, or conquest. Nonetheless, their appeal has arguably peaked, and going forward, their ability to mobilize voters and to comfortably win elections remains in doubt. Part of this has to do with their actual track record in power. Critically, it must be emphasized that most of these actors operate in authoritarian political contexts where they are constantly harassed

---

[14]  "Comprehensive Interview with Ayatullah Mesbah-Yazdi on the Efficiencies and Successes of the Islamic Revolution", *Raja News*, 29 December 2018. This interview was originally given to the hardline weekly *9 Day* and (Radio Farda 2019).

[15]  (Human Rights Watch n.d.; Euben and Zaman 2009).

[16]  (El-Affendi 1991). Also see his chapter (El-Affendi 2006).

[17]  (El-Affendi 2018).

[18]  (Rashid 2010).

and intimidated. Furthermore, they inhabit countries that face immense developmental challenges with few easy solutions. Turkey stands out as an important case study.

The Justice and Development (AK Party), under the leadership of Recep Tayyip Erdoğan, has been in power for almost 20 years. For the first decade, roughly from 2003–2013, it presided over an unprecedented period of liberalization and democratization marked by high voter turnout and substantial economic growth. During this period, Turkey's political and economic progress was widely hailed, and scholars considered it to be on the path to liberal democracy. Arab Islamists were particular enthralled by Turkey's political development. At the outset of the Arab Spring, the AK Party's blend of religion and democracy was referred to as a model that religious-based parties in the Arab world sought to emulate.[19]

The year 2013 was critical. After the civil society protests in Istanbul's Gezi Park, Erdoğan and his AK Party, having neutralized opposition from the secular and military establishment, presided over the steady de-democratization of Turkey. While signs of this were apparent much earlier, the crackdown on these protests marked a turning point. The failed military coup in 2016, blamed on the Gulen Movement, significantly enhanced the slide toward authoritarianism.[20] Using the pretext of a national security crisis, hundreds of thousands of people were arrested and fired from their jobs. Today, Turkey has the distinction of being "the biggest jailer of journalists in the world."[21] Civil society has been restricted, opposition leaders have been jailed, and President Erdoğan is rightly compared with other right wing populist leaders such as Narendra Modi, Benjamin Netanyahu, Vladimir Putin, Victor Orbán, and Donald Trump.

Erdoğan and the AK Party still have significant support within Turkey. However, in light of a declining economy and the emergence of a new breakaway party headed by Ahmet Davutoğlu (former prime minister and close Erdoğan ally), it is doubtful that the AK Party can win another national election. The mayoral election of 2019 in Istanbul, which the AK Party lost, is a harbinger of things to come.

In many Muslim societies, Erdoğan remains a popular figure. His support outside of Turkey largely stems from ignorance about the facts on the ground related to the breadth and depth of Turkey's post-2013 de-democratization. Erdoğan's popularity among Muslims is also related to his ability to manipulate powerful themes of Muslim identity related to Palestinian suffering and Islamophobia—though revealingly, not in the case of Uyghur persecution in China. In the Arab world, Erdoğan remains popular largely for these reasons, but additionally because Turkey seems more open and free compared to authoritarian regimes in North Africa and the Middle East.

Tunisia's Islamists, who now preferred to be called Muslim democrats, have the most promising track record in terms of commitment to the values and practices of liberal democracy. They played a critical role in Tunisia's transition to democracy during the Arab Spring, and they continue to play an essential role in advancing democratic consolidation. Their electoral appeal, however, has diminished, as recent election results have revealed. This has less to do with their performance in power and more to do with the dire economic conditions facing Tunisia today. To what extent the performance of the Ennahda Party under the leadership of Rached Ghannouchi remains a model and inspiration to other Islamist parties is debatable.[22]

In other parts of the Arab world, where relatively free and fair elections have taken place, the popularity of Islamists parties remains steady. Looking forward, the support enjoyed by these parties is unlikely to increase beyond current levels. Recent election results in Morocco in 2016, Lebanon in 2018, and Kuwait in 2020 would seem to bear

---

[19] (Rane 2012).

[20] (Yavuz 2021). For an excellent analysis of the slide toward authoritarianism under the AK Party see (White 2017).

[21] (Amnesty International 2017).

[22] (Al-Arian 2020).

this out. The case of Iraq deserves special mention because it clearly confirms my overall argument on the declining influence of Islamism based on their record in power.

Shia Islamist parties have dominated Iraqi politics since the removal of Saddam Hussein. Their rule has marginalized Iraq's Sunni Arab minority, but increasingly, it has also alienated Iraqi Shia Muslims, particularly the urban poor and the sizeable youth population. This was on display in October 2019, when a national uprising broke out against the Iraqi government protesting dire economic conditions, a lack of access to safe drinking water, Iran's influence on Iraqi politics, and expanding corruption. The Iraqi prime minister resigned one month after the uprising began.[23]

Most Iraqi (Shia) Islamist parties have degenerated into large patronage networks. Their priority has been using state resources to enrich their own supporters instead of focusing on national development. During the 2018 national election, the Sairoon Alliance, comprised of supporters of the populist cleric Muqtada al-Sadr and the Iraqi Communist Party, won the plurality of votes. It is noteworthy that the Sairoon Alliance campaigned on a platform of anti-corruption and on providing public services to citizens. Sectarian identity politics had lost much of its appeal, although it was still a factor. To the extent that this is a harbinger of things to come, it does not bode well for Islamist parties in Iraq and throughout the region, as economic issues and basic questions of survival take precedence for most citizens.[24]

Finally, there is ISIS. While our focus until now has been on mainstream religious-based parties and movements, the rise of the Islamic State, located at the extreme end of the political Islamist spectrum, has a bearing on how the world perceives the general phenomenon of political Islam. ISIS's primary impact has been reputational and psychological and this has also indirectly effected how Islamist parties are viewed within the region.

The rise and fall of ISIS produced massive global media coverage. Much of this pertained its extreme acts of violence, both in the Middle East and in major Western capitals. As a result, the term "Islamic State" has now been tarnished because of its association with bloodshed and barbaric rule in the name of Islam. For a long time, mainstream political Islamists aspired to create an Islamic state (*dawla al-Islamiyya*), albeit significantly different than the one Abu Bakr Al Baghdadi, the leader of ISIS, presided over. This was the explicit aspirational goal of nearly all Islamist groups in the latter half of the 20th century, as theorized and articulated by prominent Islamist leaders and intellectuals. After the rise of ISIS, the term "Islamic state" has come into disrepute. It is now associated on a global level with evil, bloodshed and barbarity. The term civil state (*dawla madiniya*) has increased in popularity among mainstream Islamists and gradually has replaced the term "Islamic state", although the origins of this term and its usage predate the rise of ISIS.[25]

## 3. Political Islam and Terrorism

There is a structural and ideological bias that has long dominated the debate on political Islam and violence. One dimension of this problem is that some acts of violence are condemned by Western governments while others are not. Typically, when a non-state actor with a religious identity engages in violence against a pro-Western target, the charge of terrorism is levied, and condemnations soon follow. By contrast, when a pro-Western regime engages in an act of violence, often with greater lethality against civilian noncombatants, the terrorism label is *not* used, and the violence is ignored, excused, or sometimes justified.[26]

---

[23]  (Human Rights Watch 2019a). Also see (Haddad 2019).

[24]  (Arraf 2021).

[25]  "The Civil State (dawla madaniya)—A New Political Term?" *IFAIR*, 24 February 2014; https://ifair.eu/2014/02/24/the-civil-state-dawla-madaniya-a-new-political-term/. Also see (Gerges 2013).

[26]  The Israel–Palestine conflict is a perfect illustration of this point. For example, during the 2008–2009 war between Israel and Hamas, human rights groups documented nearly 1400 Palestinians deaths, of which four-fifths were civilians, including 350 children. Israeli deaths were 10 soldiers (four killed by friendly fire) and three civilians. The ratio of Palestinians to Israelis killed was more than 100:1, and the ratio of Palestinian civilians to Israeli civilians killed was 400:1. See (Finkelstein 2018).

Consider the case of Henry Kissinger. In reviewing events in Egypt during the Arab Spring, he observes that after winning the presidential election in 2012, the Muslim Brotherhood "concentrated on institutionalizing its authority by looking the other way while its supporters mounted a campaign of intimidation and harassment of women, minorities and dissidents. The military's decision [in 2013] to oust this government and declare a new start to the political process was, in the end, welcomed even among the now marginalized secular, democratic element."[27]

Overlooked in Kissinger's narrative is the fact that the Egyptian military coup was accompanied by an unprecedented orgy of violence. Human Rights Watch described one particular event as a "likely crime against humanity" that may have been "the worst single-day killing of protesters in modern history."[28] The reference was to the Rab'a al-adawiya massacre on 14 August 2013, where more than 1000 peaceful protesters were killed in the center of Cairo in the span of a few hours. This event is typically ignored in mainstream Western accounts, while the categorically different and smaller-scale abuses by Mohammed Morsi are listed in detail. What is also ignored is the neo-fascist political order, backed by the West, which emerged after the coup under the leadership of the Brigadier-General Abdelfattah El-Sisi.[29] It is widely accepted that the depth and breadth of state repression in Egypt today far exceeds the darkest days of dictatorial rule under Hosni Mubarak (1981–2011). Notwithstanding this objective fact, in December 2020, France gave General El-Sisi, it highest award, the Légion d'honneur (Legion of Honor). A month earlier, Germany awarded the Egyptian Ambassador, Badr Abdelatty, the Federal Cross of Merit, its highest award.

In contrast to the regime that rules over them, the Egyptian Muslim Brotherhood has officially rejected violence as a means of obtaining political power. Even their strongest critics grudgingly acknowledge this fact.[30] Nonetheless, there is a new narrative that has emerged. Notwithstanding its rejection of violence and its commitment to electoral politics, this narrative argues that the Muslim Brotherhood still poses an ominous threat and should be ostracized. Specifically, it is claimed that the Muslim Brotherhood acts as "gateway drug" or "conveyor belt" toward radical fundamentalism in the form of Al Qaeda or ISIS. The core ideology, structure, and worldview of the Muslim Brotherhood, its critics allege, lays the foundation for an eventual and easy transition toward radical extremism and acts of terrorism.

The British government is sympathetic to this perspective. In 2015, at the request of several Arab states, it undertook an official study and review of the Muslim Brotherhood. Prime Minister David Cameron concluded that the "main findings of the review support the conclusion that membership of, association with, or influence by the Muslim Brotherhood should be considered as a possible indicator of extremism."[31] The origins of this thesis are difficult to locate.[32] In recent years, a wide chorus of opinion comprised of Arab authoritarian regimes, Western intellectuals, Islamophobic groups, and various think tanks has introduced this argument into the public intellectual and policy debate on the Middle East.

In 2015, the Egyptian government commissioned the British law firm 9 Bedford Row to prepare a report on the history, organization, and structure of the Muslim Brotherhood.

---

27 (Kissinger 2015).

28 (Human Rights Watch 2015; Human Rights Watch 2014).

29 For a general account of these events see (Kirkpatrick 2018). For a critique of the role of Egyptian liberals and secularists during and after the Arab Spring, see (Faruqi and Fahmy 2017).

30 Zachary Laub, "Backgrounder: Egypt's Muslim Brotherhood", Council on Foreign Relations, 15 August 2019, https://www.cfr.org/backgrounder/egypts-muslim-brotherhood.

31 (Government of the United Kingdom 2015).

32 Abdelwahab El-Affendi traces the origins to Egyptian intelligence sources. In the late 1970s, Islamist militants were charged with "emerging from the cloak of the Muslim Brotherhood." In the wake of 9/11, Saudi officialdom revived this claim to deflect attention away from the Wahhabi roots of the attackers. Prince Naif bin Abdulaziz, the interior minister at the time, frequently disseminated this charge. See (Al Jarallah 2002). Personal correspondence with Abdelwahab El-Affendi, 24 December 2020.

The "gateway drug" thesis is explicitly referenced in this document. "The leaders of Islamic State [ISIS], Boko Haram and al-Shabaab were first members of the Muslim Brotherhood prior to any link with the al Qa'ida network", the report states. "As a result, the Muslim Brotherhood has been described as a gateway of sorts to al-Qa'ida."[33] In a 2018 speech in London to the British think tank Policy Exchange, UAE Foreign Minister Anwar Gargash repeated this claim: "The Brotherhood is an incubator—the gateway drug—to jihadism of all kinds."[34]

In the British law firm report, the source cited to substantiate the "gateway drug" thesis is an article by Bill Roggio writing in the *Long War Journal*.[35] Mr. Roggio is a former American soldier with dubious credentials as a journalist. His contribution to this topic, which is footnoted as an authoritative reference, makes a mere passing reference to the Muslim Brotherhood. It reads more like a blog post than a serious academic contribution based on empirical research. Moreover, the *Long War Journal* is a project of the Foundation for the Defense of Democracies (FDD)—a neoconservative American think tank supportive of the policies of Donald Trump and funded by rightwing American supporters of Israel.[36]

In May 2017, the FDD organized a special conference on "Qatar and the Muslim Brotherhood's Global Affiliates." During the closing remarks, John Hannah, a senior counselor at the FDD and national security advisor to former Vice-President Dick Cheney, summarized the conference deliberations. Focusing on one theme that emerged during the day's proceedings, he observed:

> One thing that has bothered me relates to a question that my colleague Mark Dubowitz [CEO of the FDD] asked Congressman [Ed] Royce. Mark said, "But what do we do about the ideological conveyor belt?" I'd note Bin Laden, Zawahiri, even Abu Bakar al Baghdadi, the Caliph of ISIS—all of them started where ideologically? As members of the Brotherhood. It's no coincidence, I think, that the Muslim Brothers have been the gateway drug for a vast majority of violent Islamists the world over. So what do we do about that fact? Are we defenseless until they openly call for violence and start killing people? That just doesn't sit particularly well with me.[37]

While the FDD has been a leading proponent of the Muslim-Brotherhood-gateway-drug thesis, it is not the only organization to promote this claim. Other think tanks such as the Hudson Institute, the American Enterprise Institute, the Center for Security Policy, and Quilliam have done so as well.[38] In reading through their statements, a unifying theme emerges. The relationship between the Muslim Brotherhood and radicalization is rarely argued. Rather, it is mostly forcefully asserted, without any social scientific research to back it up. This is unsurprising given that most of the scholarship points in the opposite direction.[39]

This new thesis on the Muslim Brotherhood threat has also been advanced by various intellectuals. In *The Crisis of Islam: Holy War and Unholy Terror*, Bernard Lewis, writing after 9/11, argued that all Islamists groups were basically the same and did "not differ from the mainstream [of Muslims] on questions of theology and interpretation of scripture."[40] In a later interview, given during the early months of the Arab Spring, he developed this

---

[33]   (Bedford Row 2015).

[34]   (Wintour 2018). For background on the UAE and its obsession with the Muslim Brotherhood see (Cafiero 2018).

[35]   (Roggio 2010).

[36]   (Judis 2015; Clifton 2018; Foundation for Defense of Democracies 2020).

[37]   (Foundation for the Defense of Democracies 2017). Jonathan Schanzer is the Senior Vice-President of the Foundation for the Defense of Democracies. See his testimony, "The Muslim Brotherhood's Global Threat", Hearing before the Subcommittee on National Security on the Committee on Oversight and Government Reform, House of Representatives, 150 Congress, second session, 11 July 2018.

[38]   (Shariatmadari 2015) and on the Hudson Institute see, Hillel Fradkin, "The Muslim Brotherhood's Global Threat", testimony before the Subcommittee on National Security on the Committee on Oversight and Government Reform, House of Representatives, 150 Congress, second session, 11 July 2018.

[39]   (Powell 2016; Moskalenko and McCauley 2009).

[40]   (Lewis Bernard 2003).

argument further. Lewis categorically rejected the notion that the Muslim Brotherhood was "in any sense benign. I think it is a very dangerous, radical Islamic movement", he affirmed. When pressed to elaborate Lewis added: "I don't know how one could get the impression that the Muslim Brotherhood is relatively benign unless you mean relatively as compared with the Nazi party."[41]

The leading Western proponent of the Muslim Brotherhood–Nazi comparison is the American liberal intellectual Paul Berman. Drawing on the work of the historian Jeffrey Herf, he has written two widely reviewed books on this topic, *Terror and Liberalism* (Berman 2003) and *The Flight of the Intellectuals* (Berman 2010). Berman's core argument is that the Muslim Brotherhood is an ideological derivative of European totalitarian movements that emerged in the early 20th century and that the threat they pose to liberal values is very similar. Berman was particularly critical of Western liberal intellectuals for not seeing this new Muslim threat and actively opposing it. For him, this problem was embodied in the writings of Tariq Ramadan, a prominent European Muslim intellectual who is also the grandson of the founder of the Muslim Brotherhood, Hassan Al-Banna.

While acknowledging that Ramadan is not an extremist, Berman contends that his worldview acts like a gateway drug for young Muslims. Muslims exposed to the writings of Tariq Ramadan would eventually consume harder theological intoxicants. For Berman, the real Muslim reformer and hero of our time is Ayaan Hirsi Ali, the Somali-born Dutch-American feminist, author, and public intellectual. She has argued that "we are at war with Islam. And there's no middle ground in wars", adding "there comes a moment when you crush your enemy." When asked if she was referring to radical Islam, Hirsi Ali replied, "No. Islam, period." She has also suggested that Benjamin Netanyahu deserves the Nobel Peace Prize with specific reference to his policy on Gaza.[42]

Michael Walzer, the prominent American political theorist and public intellectual, has echoed these views on the Muslim Brotherhood. He credits Paul Berman as an inspiration for his thinking, specifically Berman's critique of liberal complicity with Islamism. Walzer is similarly laudatory of the moral courage demonstrated by Ayaan Hirsi Ali.[43] Hussein Ibish, a Muslim liberal intellectual who works for UAE-funded think tank, concurs with this overall assessment. "It is also accurate to compare the Brotherhood to a gateway drug for terrorism", he has written. "If only one in 10 Brotherhood members graduates to Al Qaeda, that is one too many."[44]

There is no doubt that many radical Islamists were former members of the Muslim Brotherhood. The most influential figure that comes to mind is Sayyid Qutb. Reading his earlier book *Social Justice in Islam* (Qutb 2000) and comparing it to his later incendiary book, *Milestones* (Qutb 1999), one can easily see the radical transformation and the underlying political Islamist worldview that justifies the "gateway drug" thesis. Many other prominent militant Islamists such as Abdullah Azzam and Ayman Al Zawahiri have travelled down this same road. In a trenchant critique of this argument, however, Andrew March has observed that "gates open both ways." He asks, why can't the Muslim Brotherhood "not be a gateway from orthodox Islamism to something else?"[45] Perhaps toward a more democratic and pluralist worldview.

March's point is worth considering. He draws our attention to the deep ideological and Orientalist bias that has shaped the debate on political Islam and violence. There is an inbuilt assumption that once a conservative, illiberal Islamist, always a conservative, illiberal Islamist. If there is to be any ideological transformation, it is in one direction only: toward salafi-jihadism. This is a reductive approach to the topic. It presupposes that political Islamists live in a vacuum and their ideas are not affected by local political context,

---

41 (Horovitz 2011).

42 (Berman 2003; Berman 2010). On Ayaan Hirsi Ali and Islam, see her interview (Van Bakel 2007) and also see, (Isquith 2014).

43 (Walzer 2015).

44 (Ibish 2019).

45 (March 2010).

economic shifts, demographic fluctuations, generational change, the forces of globalization, and—critically—political learning, adaptation, and evolution.

The concept of post-Islamism is relevant here. Developed by Asef Bayat in 1996, it was initially based on a set of social, ideological and political changes in post-revolutionary Iran, at the level of society, not the state. Similar developments were noticeable in other Muslim societies around the same time. The core idea of post-Islamism refers to a "metamorphosis of Islamism (in ideas, approaches, and practices) from within and without."[46] According to Bayat, post-Islamism is a multi-dimensional movement within mainstream political Islam. It refers to a "political and social condition where, following a phase of experimentation, the appeal, energy, and sources of legitimacy of Islamism are exhausted even among its once-ardent supporters." Islamists gradually develop a deeper and more nuanced understanding about the problems, inadequacies, and complexities of their societies. They undertake a process of soul searching and introspection related to state–society relations and the moral basis of legitimate political authority. Changing international factors, such as the end of the Cold War, have played a role. "Islamism becomes compelled, both by its own internal contradictions and by societal pressure, to reinvent itself, but it does so at the cost of a qualitative shift."[47]

Does post-Islamism represent an ideological change toward secularism and an abandonment of religious politics? Bayat's thesis is more nuanced. He argues that while there is a substantive ideological shift, it is not to be understood as an abandonment of religion but more a reinterpretation of it. "Post-Islamism is neither anti-Islamic nor un-Islamic or secular", he observes.

> Rather, it represents an endeavor to fuse religiosity and rights, faith and freedom, Islam and liberty. It is an attempt to turn the underlying principles of Islamism on their head by emphasizing rights instead of duties, plurality in place of a singular authoritative voice, historicity rather than fixed scriptures, and the future instead of the past. It wants to marry Islam with individual choice and freedom (albeit at varying degrees), with democracy and modernity, to achieve what some have termed an 'alternative modernity'.[48]

Early examples of post-Islamist parties include the reform movement in Iran in the late 1990s, Indonesia's Prosperous Justice Party, Malaysia's People's Justice Party, Egypt's Center Party (*Hizb al-Wasat*), Morocco's Justice and Development Party (PJD), and Turkey's Justice and Development Party (AKP) prior to 2013. Each of these groups emerged from the mainstream Islamist current in their country yet came to rethink the relationship between religion, state and society over time. The model of Christian Democratic Parties in Europe most closely approximates this transformation.[49]

## 4. A Firewall, Not a Gateway Drug

There is a longstanding and widely recognized inverse relationship between democratic societies and violence. The more that democracy advances, in the form of political accountability, public transparency, and the peaceful transfer of power, the less the likelihood of violence. This is a variation of the famous democratic peace theory. It draws upon a long intellectual tradition going back to the Enlightenment where thinkers such as Immanuel Kant, Jean-Jacques Rousseau, Thomas Paine, and Alexis de Tocqueville contributed to an argument that democratic societies are more internally peaceful and less likely to go to war than their non-democratic counterparts.[50] This theory is pertinent to understanding the relationship between political Islam and violence.

---

[46]  (Bayat 2013).

[47]  Ibid., p. 8.

[48]  Ibid.

[49]  (March 2015).

[50]  According to George Kateb (1961), the goal of Rousseau's democratic project is justice.

Observing the general turmoil in the Middle East today, it is easy to forget that a few years ago, the region looked quite different. The 2011 Arab Spring brought hope to people of the region that a brighter political future might be possible. Starting in North Africa and moving in quick succession toward the shores of the Persian Gulf, pro-democracy revolts swept three longstanding dictators from power and came close to removing another two. In the streets of many Arab capitals, a unifying slogan that encapsulated the demands of a new generation of Arabs: *ash-shab yurid isqat an-nizam* (the people want to bring down the regime). These events shook the foundations of Middle Eastern authoritarianism while capturing the imagination of the entire world.

Al Qaeda's response to the Arab Spring was revealing. It was stunned by these revolts, which produced ideological confusion and organizational incoherence. In an important study, *Jihadi Discourse in the Wake of the Arab Spring*,[51] the authors noted that during the Arab Spring, salafi-jihadi groups were both impotent and unpopular. The reasons for this are self-evident. During the Arab Spring, it appeared that political change could be achieved via peaceful protest rather than by violent revolution. This undermined one of the central ideological claims of Al-Qaeda, who have long argued that dictators could only be removed via armed struggle; democratic elections and nonviolent protests could never work. As Ayman Al-Zawahiri famously put it, "what is truly regrettable is the rallying of thousands of duped Muslim youth in voter queues before ballot boxes instead of lining them up to fight in the cause of Allah."[52] As a result, the ideological appeal of Islamist militancy during this period fell precipitously throughout the Arab-Islamic world.

The Arab Spring, however, was rolled back and ultimately defeated. A counter-revolution led by the previous ruling elites, the deep state, acquiescence from the West, and—critically—strong support from Saudi Arabia and the United Arab Emirates all contributed to this outcome.[53] The promise of peaceful political change and the doors to democratization were firmly slammed shut. As a direct result, there has been an increase in violence and extremism across the region. This demonstrates another important relationship that is central to the politics of the Middle East: when democratic openings are closed and moderate forms of political Islam are crushed, radical Islam and violence thrive as a consequence. Egypt is a case in point.

In the 22-month period since Egypt's July 2013 coup, there were more than 700 acts of violence across Egypt compared to 90 attacks in the previous 22 months.[54] This lower figure overlaps with a democratization process that was in place in Egypt following the ouster of Hosni Mubarak. The huge spike in violence after the military coup is not a coincidence either. Since then, Human Rights Watch has reported the arrest and detainment of approximately 60,000 political prisoners (mostly members of the Muslim Brotherhood), many of whom have been tortured. According to Amnesty International, Egypt is an "open air prison for critics", and as 2020 came to a close "Egyptian authorities . . . embarked on a horrifying execution spree . . . putting scores of people to death, in some cases following grossly unfair mass trials", according to Philip Luther, Amnesty International's Middle East and North Africa Research and Advocacy Director.[55]

The number of young people radicalized by these events is difficult to measure. To the extent that anecdotal evidence, media reports, and trends on social media are a reflection of this tendency, it is accurate to state that Egypt has become a breeding ground for radical Islamism. Marc Lynch has argued that, notwithstanding the Muslim Brotherhood's social conservatism, they have acted as a "firewall" against extremism.[56] The organization is able to "'capture' Islamists within a relatively moderate and peaceful movement and

---

51　(Lahoud 2013).

52　(El-Ghobashy 2005).

53　(Peel and Hall 2013). Also see (Filiu 2015; Steinberg 2014).

54　(Kagan and Dunne 2015). Also see (Tahrir Institute for Middle East Policy 2018).

55　(Human Rights Watch 2019b; Amnesty International 2018, 2020b).

56　(Lynch 2010).

[this] prevents their evolution into more radical, violent actors." When viewed from the perspective of politics on the ground in a deeply repressive state, Muslim Brotherhood members "are more likely to remain committed to the MB's methods and doctrines, and to be more able to resist the temptations of the radical path to *jihad*."[57]

Before they were officially banned, Egyptians with a religious identity could find expression in the public sphere by joining the Muslim Brotherhood. They could participate in civil society organizations and in electoral politics. Since the 2013 coup and the attempt to eradicate the Muslim Brotherhood, this option no longer exists. There are only two options open to Egyptians today: (1) to remain silent and accept the current neo-fascist order or (2) to contemplate joining a utopian revolutionary political project such as ISIS. There is no third alternative.[58] Tales from Egypt's notorious prison system confirm this argument.

Mohammad Soltan, an Egyptian-American, was 25 years old when he was arrested in the summer of 2013. He spent 21 months in jail. During 16 of these months, Soltan was on a hunger strike. He lost 160 pounds, risking organ failure. When he emerged from prison, he could not walk. In a special *New York Times* profile, he discussed the torture and brutality he faced, but also revealed details of the internal political debates among prisoners. Several of his cellmates were ISIS supporters.[59]

"They walked around with a victorious air", he recalled. They would frequently point to supporters of the Muslim Brotherhood and state: "look, you idiots, your model doesn't work." The ISIS supporters would then proceed to "make very simple arguments telling us that the world doesn't care about [democratic] values and only understands violence." He also noted that because "of the gravity of the situation [we] were all in, by the time the ISIS guys were finished speaking, everyone, the liberals, the Brotherhood people, would be left completely speechless. When you're in that type of situation and don't have many options left, for some people these kinds of ideas start to make sense."[60]

Tunisia, the one Arab Spring country that did undergo a successful democratic transition, provides an alternative model to that of Egypt. Rached Ghannouchi, the leader of Ennahda and current speaker of parliament, has observed that the "only way to truly defeat ISIS is to offer a better product to the millions of young Muslims in the world." It is called "Muslim democracy." He noted that most "young people don't like ISIS—see how many millions flee from it—but they won't accept life under tyrants either." This "better product" must be a political system that is democratic, that respects human rights, and that gives Islamic values political space.[61]

It is not a coincidence that ISIS emerged and attracted followers *after* the crushing of the Arab Spring. This fact highlights the relationship between democratization and violence. The simplest formulation of this insight into modern politics was best articulated by John F. Kennedy in 1962: "Those who make peaceful revolution impossible will make violent revolution inevitable."[62]

## 5. Conclusions

Approximately 40 years ago, political Islam burst onto the world stage. Seemingly, it has never left. Different manifestations of this phenomenon have continued to generate mostly negative headlines, often for good reasons. This article has sought to enhance analytical clarity by approaching this topic retrospectively. Two separate but interrelated themes were examined: the legacy of political Islam in power and the relationship between political Islam and violence.

---

[57]    Ibid., pp. 468, 480.

[58]    (Daragahi 2014; Shahin 2015).

[59]    (Kirkpatrick 2015b).

[60]    (Shackle 2015; Hussain 2015).

[61]    (Zakaria 2015). Also see (Ghannouchi 2016).

[62]    (Kennedy 1962).

It was argued that, all things considered, the various experiments of political Islam's acquisition of state power, has reflected negatively on this socio-political movement. This claim applies especially to those cases where Islamists came to power via revolution (Iran), a military coup (Sudan), or through military conquest (Afghanistan). Examples of Islamism's acquisition of state power via the ballot box in Turkey and different parts of the Arab world has not substantially elevated the popularity of religious-based parties. Arguably, their appeal has peaked, and at best, Islamists will be able to retain their current level of support, which is unlikely to increase in the coming years. While the reasons for this are many, poor job performance, as I have argued, is a critical factor in explaining this outcome.

Measuring the popularity and prestige of political Islam is difficult today. In the aftermath of the Arab Spring, several Arab countries (Egypt, Jordan, Saudi Arabia, Bahrain, and the United Arab Emirates) have officially banned the Muslim Brotherhood or declared them a terrorist organization. A vicious crackdown and a misinformation campaign have been launched to discredit the Brotherhood in the court of global public opinion. This state-sanctioned repression is among the worst the region has experienced since independence. According to Amnesty International, in 2019, "excluding China, 86% of all reported executions [in the world] took place in just four countries—Iran, Saudi Arabia, Iraq, and Egypt." This has contributed to giving the Middle East a unique status—it is the most politically repressed part of the world, according to the 2020 Human Freedom Index.[63]

The debate on political Islam and violence was also scrutinized in this article. An old controversy has resurfaced in recent years: is the Muslim Brotherhood a "gateway drug" toward radicalization or does it act as a firewall by restraining violent activity? The latter proposition was defended. It was argued that advocates of the Muslim-Brotherhood-gateway-drug thesis rarely cite empirical studies to substantiate their claim. Most proponents of this critique are politically motivated in the sense that their views on this subject are a consequence of their support for the ruling elites of various Arab states or due to their affinity with Israel, which now is openly allied with these authoritarian regimes. The Muslim Brotherhood is thus criticized not because they are a purveyor of violence but fundamentally because they oppose the regional status quo and the political regimes that seek to maintain it.

In the introduction to the *Princeton Readings in Islamist Thought*, the editors make an important observation. They suggest the violence some "Islamists proffer mirrors the very state-sanctioned violence against which they have struggled for almost a century."[64] Many of the most prominent advocates of violent revolution in the Arab-Islamic world, from Sayyid Qutb to Abu Musab al-Zarkawi to Ayman al-Zawahiri, are products of prison systems in the Arab-Islamic world where they have spent years in jail subject to unspeakable cruelty. It is unsurprising, therefore, that people exposed to prolonged torture and extreme interrogation conclude that violence is a legitimate political tool. According the Palestinian Islamist Khaled Abu Hilal, "prison is my university", a point that was also eloquently made a hundred years ago by the Russian author and Marxist revolutionary Maxim Gorky.

> A people brought up in a school that reminds one of the torments of hell on a small scale, a people accustomed to the clenched-fist, prison, and the whip, will not be blest with a tender heart. A people that the police agents have ridden over will be capable in their turn of walking over the bodies of others. In a country where unrest has reigned so long, it is difficult for the people to realize from one day to the next the power of right. One cannot demand from a man who has never known justice that he should be just.[65]

---

[63] (Vasquez and McMahon 2020; Amnesty International 2020a).

[64] (Euben and Zaman 2009).

[65] Ibid., p. 44.

This draws our attention to the structural conditions in Arab-Islamic world that produce violence. This analytical approach is often ignored when the question of political Islam and violence is discussed. In this context, what is truly fascinating and in need of explanation are not the acts of violence produced by some non-state actors in the Middle East but why this region has not produced more violence by these groups in light of the existing repressive socio-political conditions. This approach to the question of political Islam and violence is rarely pursued and would benefit from further research by scholars and social scientists.

**Funding:** This research received no external funding.

**Institutional Review Board Statement:** Not applicable.

**Informed Consent Statement:** Not applicable.

**Data Availability Statement:** Not applicable.

**Conflicts of Interest:** The author declares no conflict of interest.

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
