# Peer review of "Political Islam: A 40 Year Retrospective"

_religions, doi:10.3390/rel12020130_

Round 1

Reviewer 1 Report

I commend the author on this article. It provides a broad overview of political Islam over the past 40 years, including the fate of Islamist experiments in the eyes of Muslim populations as well as the arguments concerning the relationship between political Islam and violence. The article is well-structured, well-written, and its observations are generally well supported. I can recommend only minor revisions:

  • Line 48: the word "often" could be added before the mention of political Islam being "erroneously equated with violence...." as the association is often not erroneous.
  • Line 58: some further details could be provided in relation to support for political Islam in the Muslim world. In addition to the reference to Arab Barometer, the author could reference studies on the popularity of Islamists among Muslims e.g. Kurzman, C. & Naqvi, I. (2010). Do Muslims Vote Islamic? Journal of Democracy, 21(2), 50-63.
  • Line 65: some details and references could be provided in relation to the "political theory" developed by political Islamists. Readers would benefit from the author's insights on the similarities and differences of political theory(ies) expressed by such groups/parties as Iran's ruling regime, the Taliban, Muslim Brotherhood, Hizbut Tahrir, Turkey's AKP, etc.
  • Line 67: in relation to Line 58, some further details about the appeal of political Islam would be useful here.
  • Line 182-183: add a reference here in relation to the Turkish model and the Arab world e.g. Rane, H. (2012). An Alternative to the" Turkish Model" for the Emerging Arab Democracies. Insight Turkey, 14(4).
  • Line 210: add the word "role" and delete "the"..."They played a critical [role] in [the] Tunisia's transition to democracy..."
  • Line 257: subheading for Political Islam and Terrorism needs to shift up a level.
  • Line 462: add the word "a"...."As [a] result..."

A general copyedit is recommended to correct additional typos.

Author Response

Since reviewer 1 report was all minor revisions, mostly related to typos and with a request to add a few more references, I have accepted all of them and made these changes via track changes. 

Reviewer 2 Report

The aim is to present an overview of Islamism the last 40 years, and the main material used is secondary. In one sense, nothing is really new in the article, but it brings forth the main views and scholarly discussions related to Islamism and is a valuable overview article, that could gain a broader interest and also function in a variety of courses on political Islam. The presentation is also nuanced and gives many examples.

I think that the author should attempt to comment on the terminology a bit more. For example, “political Islamism” is used. What is that compared to “Islamism”. The examples that are presented range from a wide variety of countries, regions and Islamic affiliations, so I think that a discussion of the term “Islamism” would be good, and also if possible a comment on why these examples are chosen. Is it the MENA-region? No examples of the MB in Europe or other political groups in Asia for example are mentioned. This means that I wish for a brief comment on the choice of examples to illustrate the discussion.

Author Response

This reviewer asked me clarify the use of terminology related to the terms "political Islam" and "Islamism." I have done so in footnote 1 and I believe this addresses the concerns of this reviewer. See the changes I made in track changes.